# Performance Evaluation and Cyberattack Mitigation in a Blockchain-Enabled Peer-to-Peer Energy Trading Framework

**DOI:** 10.3390/s23020670

**Published:** 2023-01-06

**Authors:** Nihar Ranjan Pradhan, Akhilendra Pratap Singh, S. V. Sudha, K Hemanth Kumar Reddy, Diptendu Sinha Roy

**Affiliations:** 1School of Computer Science and Engineering, VIT-AP University, Amaravati 522237, India; 2Department of Computer Science and Engineering, National Institute of Technology Meghalaya, Shillong 793003, India

**Keywords:** blockchain, distributed ledger technology (DLT), peer-to-peer energy trading, cyberattack mitigation

## Abstract

With the electric power grid experiencing a rapid shift to the smart grid paradigm over a deregulated energy market, Internet of Things (IoT)-based solutions are gaining prominence, and innovative peer-to-peer (P2P) energy trading at a micro level is being deployed. Such advancement, however, leaves traditional security models vulnerable and paves the path for blockchain, a distributed ledger technology (DLT), with its decentralized, open, and transparency characteristics as a viable alternative. However, due to deregulation in energy trading markets, most of the prototype resilience regarding cybersecurity attack, performance and scalability of transaction broadcasting, and its direct impact on overall performances and attacks are required to be supported, which becomes a performance bottleneck with existing blockchain solutions such as Hyperledger, Ethereum, and so on. In this paper, we design a novel permissioned Corda framework for P2P energy trading peers that not only mitigates a new class of cyberattacks, i.e., delay trading (or discard), but also disseminates the transactions in a optimized propagation time, resulting in a fair transaction distribution. Sharing transactions in a permissioned R3 Corda blockchain framework is handled by the Advanced Message Queuing Protocol (AMQP) and transport layer security (TLS). The unique contribution of this paper lies in the use of an optimized CPU and JVM heap memory scenario analysis with P2P metric in addition to a far more realistic multihosted testbed for the performance analysis. The average latencies measured are 22 ms and 51 ms for sending and receiving messages. We compare the throughput by varying different types of flow such as energy request, request + pay, transfer, multiple notary, sender, receiver, and single notary. In the proposed framework, request is an energy asset that is based on payment state and contract in the P2P energy trading module, so in request flow, only one node with no notary appears on the vault of the node.Energy request + pay flow interaction deals with two nodes, such as producer and consumer, to deal with request and transfer of asset ownership with the help of a notary. Request + repeated pay flow request, on node A and repeatedly transfers a fraction of energy asset state to another node, B, through a notary.

## 1. Introduction

There has been a rapid global population growth during the past few decades. Some even go so far as to call electricity a “necessity” for human survival. In order to strike a balance between supply and demand, we have relied on conventional power plants such as those that run on fossil fuels up to this point. Reducing reliance on fossil fuels, restricting grid expansion, and bolstering innovative technology are all pressing issues in the energy business. Thus, the electricity supply, transmission, and distribution sectors have had to deal with rising consumer demand throughout the years [1,2,3,4]. To balance the rising demand and limited supply, renewable energy sources such as solar are rising to the forefront. Keeping track of all the many players, utilities, manufacturers, customers, etc., adds another layer of complexity. Centralized ledgers have traditionally been used to record transactions between players in the energy market; but, as the market evolves, these ledgers have become increasingly inefficient, slow, cumbersome, and expensive to use. Because of this, numerous tasks, such as monitoring energy demand and supply, ensuring the safety of producers and consumers, calculating costs, and settling payments quickly, necessitate a substantial investment of time and energy [3,5]. Sometimes, security breaches can occur due to a lack of responsibility in the power industry, which can lead to a number of different types of errors [6,7,8]. The issues in peer to peer energy trading system, from generation to transmission to distribution, are shown in Figure 1.

It shows the issues related to energy trading process, starting from producer generation to consumer utilization. It also represents how blockchain is a feasible solution to handle all these issues.

Blockchain’s inherent characteristics, such as decentralized platform, transparency, auditable, irrevocable digital ledger, etc., attract many organizations to adopt it [9,10,11]. The viable solution to future energy, which needs the system to be secure, efficient, decentralized with respect to energy records, digitized with respect to technologies, democratized with respect to more consumer participation, and decarbonized with respect to carbon free green energy resources, is integration of blockchain technology [12,13,14]. A distributed energy system using blockchain technology can help due to its novel characteristics and can manage the energy transaction efficiently in a real-time problem. Despite all these impressive advantages, this technology faces many inherent challenges, such as transparency, security, privacy, and low scalability. To overcome the abovementioned challenges, tremendous research efforts have been underway toward a new paradigm, such as the Corda and Hyperledger frameworks. Therefore, in this paper, a blockchain-enabled prototype is designed and implemented for a peer-to-peer energy trading framework using the Corda network notary services that disseminate fair transaction distribution and reduce the effect of delay trading cyberattacks. This also helps in maintaining transparency, security, and privacy among the actors involved in this energy trading, thus preventing any form of miscommunication.

The major contributions of this paper include the following:The blockchain-enabled peer-to-peer energy trading framework implementation and prototype design is presented.R3 Corda is used for enacting smart contracts for client communications. A thorough performance evaluation of this prototype is presented herein.A novel class of cyberattacks in energy trading, such as delay trading and discard, is introduced. We design a threat-model, adversary effect, and mitigation of these attacks. The double-spending attack in the proposed energy trading process is also mitigated through the presence of the notary in the network.We also develop, analyze various smart contracts, deploy nodes, state test, signer test, and transfer command contract rules in order to handle a novel peer-to-peer energy trading process in the proposed framework.We carry out measurement and benchmarking of performance parameters such as message rate and flows, metering, send and receive rate, throughput, JVM heap memory usage, and latency, by using Grafana visualization tool for the proposed P2PET.The framework provides a confidential identity security to all the trading participants by using network map services. The identities are only distributed to other participants on a need-to-know basis. We integrate use of the latest and far more reliable transaction broadcasting and validation services such as notary, attachment, and network map services.

This papers serves as a guideline and presents a complete and comprehensive performance and cyberattacks study for blockchain-based energy trading systems, with state-of-the-art Corda DLT network schemes not yet investigated in the literature. The organization of the paper is as follows. Section 2 briefly discusses the related work. The proposed Corda-based peer-to-peer energy trading framework system architecture and modeling to mitigate the cyberattacks are presented in Section 3. Section 3 describes the deployment and implementation of the framework. Based on the analysis of the performances, discussion is given in Section 5. Finally, a brief conclusion is presented in Section 6.

## 2. Literature Review

Only a small number of publications have addressed the topic of blockchain benchmarking as a whole, and much less the P2P energy trading process in particular. In addition to Fabric and IOTA, Esmat et al. [15] also used another layer of blockchain to provide security through smart contracts and outlined an ant colony optimization approach for stakeholders in the energy market. For instance, Hassija et al. [16] developed a token-based energy trading system for UAVs and charging stations that is based on the distributed ledger technology IOTA Tangle. IOTA addressing reuse in distributed ledger technology was also evaluated by Shafeeq et al. [17] using a cuckoo filter. Using Hyperledger Fabric and Composer, the creators of [2] created a blockchain-enabled multiparty healthcare platform. Furthermore, they created participation access criteria and utilized Hyperledger Caliper to quantify the outcomes. As a counterexample, Park et al. [18] investigated the viability of an online marketplace for trading energy using a distributed ledger (DAG) that does not rely on blocks.

Various distributed ledger technology (DLT) platforms, including R3 Corda, Hyperledger Fabric, Sawtooth, Burrow, Ethereum, IOTA, etc., were analyzed and compared by Chowdhury et al. [2]. A variety of efficiency indicators were chosen by the writers. Researchers discovered that the R3 Corda network not only has an extremely low energy footprint, but also protects user anonymity and confidentiality. They also demonstrated that the scalability of the Corda network’s transactions is high because only relevant nodes are involved and a notary and other services are utilized. When compared to other blockchain frameworks, the measured performance is exceptional. Unified Modeling Language (UML), created by Gorski et al. [4], can be used with Corda’s distributed ledger technology. They also came up with the categorizations and weights for the attributes used in the DLTs. UML deployment and the Gradle Groovy script were the starting and ending points of the aforementioned implementation. Most of the work related to blockchain in energy trading is based on Ethereum Smart Contracts, for example, the work presented by Want et al. [19]. Though Ethereum makes it easier to develop any kind of decentralized application based on smart contract, it uses permissionless mode of operation. In a permissionless network, any number of nodes can join the network at any time, which slows down network computing over the time and makes the network less transparent. Total transparency comes at the cost of scalability and privacy. Few authors have presented work related to cyberattacks in a peer-to-peer energy trading system. For example, Wang et al. [20] discussed the role of blockchain in energy trading and mitigation of cyberattacks, and talked about the terms such as digital access rules and data immutability. Pang et al. [21] gave a survey in detail on recent developments in the security of NCSs deception attacks from IT and system control, respectively. The authors discussed security incidents reported in recent years and reviewed a couple of prevailing cyberattacks. Table 1 presents work on blockchain-based peer-to-peer energy trading systems. Even though most of the studies employed Hyperledger Platform or Ethereum virtual machines, the transaction broadcasting facility of the network has a relatively low speed for both. In addition, the metrics such as P2P, metering, flow rate, heap memory usages, etc., are rarely covered in depth in the literature. Similarly, research into the reliability and robustness of the R3 Corda DLT is still in its infancy.

## 3. System Architecture for Proposed Framework

Deploying the peer-to-peer energy trading framework on a Corda network with multiple distributed ledger technology (DLT) nodes, such as energy producer as party A, consumer as party B, notary, and network map node, provides interoperability of public networks with the privacy of a private network. The network model, ledger, energy states, transactions, time window, flows, attachment, and contracts are discussed to justify the applicability of the proposed framework. For our proposed peer-to-peer energy trading process, we consider two types of loads: unresponsive and responsive load. Unresponsive loads can be defined as that the consumption of energy does not change with respective to varying prices. Responsive load is one where the consumption varies because of heating, ventilation, and air conditioning (HVAC) and the consumers adjust their loads according to the price. The energy bidding in our framework works with the following tuples in Equation (1).
(1)Bid=<λ,α,β>
where λ represents time window of Corda for delivery, α is available maximum energy, β is the minimum or maximum price reserved.
(2)$$s_{i}\left(b_{i},p\right)=t_{i}\left(b_{i}\right)-b_{i}.P$$
(3)$$s_{j}\left(b_{j},p\right)=b_{j}.p-C_{j}\left(b_{j}\right)$$

We considered the set of consumer as *C*, set of producer as *P*, and consumer usage iϵC. $t_{i}\left(b_{i}\right)$ represents the consumer benefits for consuming $b_{i}\geq 0$ energy, and łp is the unit price. Similarly, the energy producer jϵP depends on its produced energy minus generating cost, as shown in Equation (3). $C_{j}\left(b_{j}\right)$ benefits for producing $b_{j}\geq 0$. In our proposed work, the maximum benefit for both the producer and consumer is considered, as shown in Equation (4).
(4)$$f\left(b,p\right)=\sum_{i\epsilon 1}^{C}s_{i}\left(b_{i},p\right)+\sum_{j\epsilon 1}^{P}s_{j}\left(b_{j},p\right)$$

The demand supply balance in the energy market is maintained by equating the amount of energy produced and sold, as shown in Equation (5).
(5)$$\sum_{i\epsilon 1}^{C}b_{i}=\sum_{j\epsilon 1}^{P}b_{j}$$

Formulating the resource allocation and maximizing the benefits of both the parties, we generate a solution by an optimizing Equation (6).
(6)$$Max_{b,p}f\left(b^{*},p^{*}\right)=\sum_{i\epsilon 1}^{C}b_{i}=\sum_{j\epsilon 1}^{P}b_{j}$$

We consider a threat model where the adversary’s goal is to maximize their profits. The producer has an intention to maximize the price while minimizing the energy amount. Similarly, the consumer has an intention to minimize the price against maximizing energy consumption. We only considered the adversary producer who either delays (or discards) the bids of the consumer and transfers the energy with an market equilibrium $b_{j}^{a},p^{a}$ and thereby increases the profit by π, as shown in Equation (7). *AP* denotes the adversary producer.
(7)$$\sum_{j\epsilon AP}^{}s_{j}\left(b_{j}^{a},p^{a}\right)=\sum_{j\epsilon AP}^{}\left(1+\pi \right)s_{j}\left(b_{j}^{*},p^{*}\right)$$

The produced energy amount before and after cyberattacks are shown in Equations (8) and (9).
(8)$$Q^{*}=\sum_{j\epsilon AP}b_{j}^{*}After_{attack}$$
(9)$$Q^{a}=\sum_{j\epsilon AP}b_{j}^{a}Before_{attack}$$

Now, aggregating the cost function and solving with an quadratic function, we obtain Equation (10), where $\sigma_{2}$ and $\sigma_{1}$ are constants.
(10)$$\sum_{j\epsilon P}^{}C_{j}\left(b_{j}\right)\approx C\left(Q\right)=\sigma_{2}.Q^{2}+\sigma_{1}.Q$$

The producer without an attack will be given in Equation (11).
(11)$$\sum_{j\epsilon P}^{}s_{j}\left(b_{j}^{*},p^{*}\right)=Q^{*}.p^{*}-C\left(Q^{*}\right)$$

Now, substituting the value of $P^{*}$ = $C\left(Q^{*}\right)$, we obtain
(12)$$\sum_{j\epsilon P}^{}s_{j}\left(b_{j}^{*},p^{*}\right)=\sigma_{2}{\left(Q^{*}\right)}^{2}$$

Again, substituting Equation (12) in Equation (10), we obtain
(13)$$Q^{a}=\left(\sqrt{1+\pi }\right)Q^{*}$$

The total energy increase in market by an adversary producer is shown in Equation (14).
(14)$$\Delta Q^{a}=Q^{*}\left(\sqrt{1+\pi }-1\right)$$

### 3.1. Proposed Network Model

In this section, a *Corda* blockchain-based P2P energy trading application is proposed on various DLT nodes with *notary, signer permissioning, confidential identities in network map node, and energy trading contract to handle request, transfer commands*, as depicted in Figure 2. For the proposed framework on a Corda network, we consider two parties, producer (trader A), consumer (party B) and matching notary. The matching notary is one to whom the energy will be requested from party A and then it transfers to party B after verifying the trading rules and data in the *energy contract* and *attachment*. To achieve this we consider an *energy state* and an energy contract and a flow called requestEnergy flow. Once the party A requests energy, we can have another flow, called transfer energy flow, to party B. We can have any number of producers and consumers on the network and they can transfer energy state to each other back and forth as many times as they like. We considered number of attributes on energy state such as energy type, demand and supply in KW, time of delivery, price per KW, issuer, and owner. For an issuance transaction it will have no input, one output, it only accepts if energy is solar or wind, and a producer signature. Thus, when the flow is completed we would have the energy state in the producer’s, notary’s, and consumer’s vaults. Now that the producer is owner of the energy state, he would be able to transfer energy flow, which takes the output of the previous transaction as input for this transaction and sends it to consumer. Though it is a transfer energy transaction flow, both consumer and matching notary sign the transactions. The energy contract has separate rules for transfer command. Once the transfer energy flow is completed, we received the energy state being used as input to be marked as *consumed* in both producer and notary vault, and a new energy state is created in the consumer vault, which will be *unconsumed*, and the owner for that energy will be the consumer. Similarly, the payment state and transactions are designed in the framework. The other entities in the proposed network are discussed below.
*Nodes and Vaults*: *Corda* network is a permissioned peer-to-peer network of nodes that accesses control by a doorman. Each producer and consumer node runs the Corda software as well as the Corda applications, known as *Cordapps*. Each node in the network maintains a vault. The vaults also maintain the *consumed* and *unconsumed* energy states.*Node Identities*: Each node has an well-known identity which is used to represent the transaction. We designed two types of identities: legal and service identities. Legal identities are used for the producer and consumer transactions. Service identities are used for those providing transaction-related services, such as *notaries* and *Oracle*. Our proposed framework is designed to generate confidential identities for nodes for individual transactions.*Network Map Service*: The network map service in the proposed network publishes the IP addresses of all the participants in the energy trading process. The consumer may query about their demand of energy; similarly, the producer may write their supply of excess energy. All the nodes in the network can be reached with the identity certificate from network permissioning services. This certificate guarantees the node identities while communicating with other peers in the network. It generates confidential identities that are not published in network map services. It ensures the transaction participants’ identities even if an intruder attacker obtains access to the unencrypted transaction. The chain linking to confidential identity is disclosed only on a need-to-know basis.

### 3.2. Proposed Ledger, States, Transactions, Time Window, Notary, and Contract

For the proposed blockchain-based P2P energy trading framework, the basic entities are ledger, states, transactions, time window, notary, and contract, designed through Corda. These entities are designed to perform some particular functionality governed by a set of rules called smart contracts.
Ledger: The participants of our proposed framework maintain separate databases of facts or states in their ledgers shared with everyone if they need it. However, unlike Bitcoin, the transactions are not globally broadcasted. Therefore, we share facts or states with the participants on a need-to-know basis. When peers make transactions, their respective ledger is updated only. Considering an instance, if person X makes 3 KW of energy trading to Y and makes an entry into the Corda network about an energy transaction, then both X’s and Y’s ledgers receive updates but nobody else in the network is allowed to know about the transaction.States: It is defined as an immutable object that represents the facts from one or more peers from the Corda network, at a particular point of time. These are differentiated by marking the current state as historic and creating an updated state. It also contains states such as consumer states as historic data and current states as nonconsumed data. It also marks a timestamp along with the states. The state objects are stored in a vault from time to time.Transactions: The first transaction would have zero inputs, i.e., X does not have energy in the DLT. X is issuing energy, so there is no input state and one output state in the first transaction. Since producer <X1,X2,X3,…> and consumer <Y1,Y2,Y3,…> are both participants in the transaction, they both need to input their signature, as shown in Equation (15).
(15)$$SIG_{X1}+SIG_{Y1}\to TX1_{Energy}$$After one or two transactions, the vault represents the unconsumed and consumed energy amount along with the payment state. The payment state needs to be signed by the consumer, as shown in Equation (16).
(16)$$SIG_{Y1}\to TX2_{Payment}$$For each transaction, a unique transaction hash is generated. Index is the count of the state.Attachment: Attachments used in the framework contain a large set of data that can be used across different transactions. The attachment contains historic trading data, table of prices for different qualities of produced energy, such as solar and wind, maximum and minimum prices, etc. Each transaction can refer to zero or more attachments. The information in these attachments can then be used to validate the transaction.Commands: Including a command in a transaction allows it to indicate the transaction’s intent, affecting how we change the validity of the transactions. Settle command is used to settle the producer’s and consumer’s energy amounts by signing both of them. Similarly, the pay command is used and signed by the consumer as they are the owner of the cash, shown in Equation (16).Time Window: In our proposed peer-to-peer energy trading framework, the transfer and payment settlement transaction need to be performed in a certain timeframe. Thus, the time window in our network represents a transaction commit validation before or after a particular time, or within a particular timeframe. For our application, we used from and to date along with time duration for trading.Notary: The transaction uniqueness consensus is achieved through notary network services. Basically, this consensus solves the problem of double-spending attack in a peer-to-peer energy trading process, because it signs a particular transaction that consumes any of the transaction state of the proposed system. The transaction and system finality is achieved by this. Without a notary, the trading transaction can be neither finalized nor notarized.
(17)$$SIG_{X1}+SIG_{Y1}+SIG_{Notary}\to TX3$$Contract: A transaction is valid if it is digitally signed by all required signers and if it is contractually valid. In the proposed work, the energy state requires an energy contract, which have certain rules as described in Algorithm 1. We used two algorithms for the proposed framework. Algorithm 1 describes the peer-to-peer energy trading process in a Corda network where the energy producer and consumer can process their demand and request through an API. Algorithm 2 performs a matching in the Corda network by comparing the digital signature of each party, demand from producer, and request from consumer. Algorithm 2 performs energy matching from the consumer and producer by a notary available in the Corda network. Similarly, the payment state would also refer to payment contract.Flows: A flow is a sequence of steps that represents a node behavior to achieve a specific ledger update. It also automates the process of agreeing on ledger updates. Automatic common tasks can also be provided with the help of a builderin flow. Figure 3 represents the flows in our proposed framework. Request energy and pay cash are sub-flows used in the proposed framework. A flow that started as a sub-process is known as a sub-flow.
**Algorithm 1:** Peer-to-peer energy trading in Corda network.
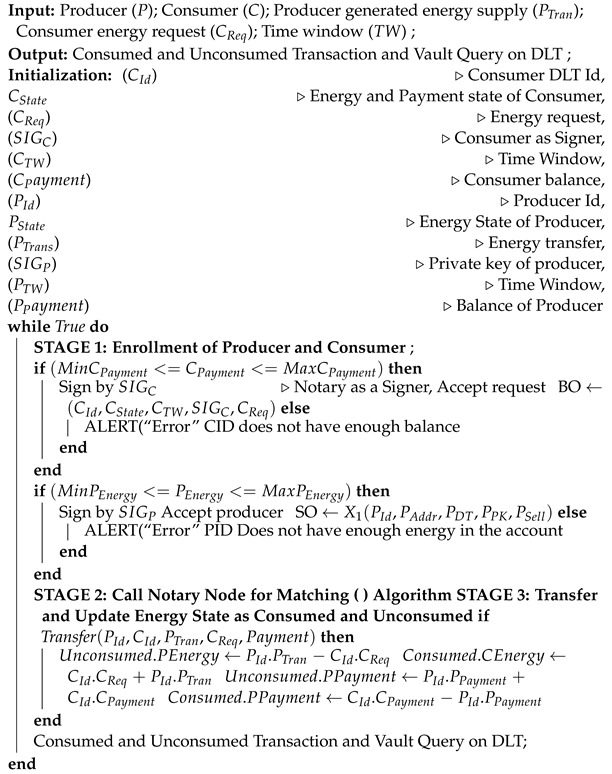

**Algorithm 2:** Notary validation by energy matching attachment.
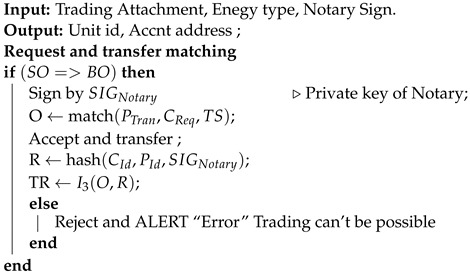


## 4. Threat Model

This section describes a threat model with respect to the adversary and the attacker.

### 4.1. Attacking Capabilities of the Adversary

We presume that the adversary is unable to alter or remove bids that have been accepted by the market and that it is unable to disrupt the market-clearing mechanism (the blockchain guarantees that this would require significant resources). An adversary, who may be a prosumer themselves, can view historical bids and clearing prices stored in the blockchain. Cryptocurrency and public key thefts can occur due to flaws in blockchains. These vulnerabilities could be exploited by an adversary to manipulate the offers made by prosumers. An attacker could, for instance, steal prosumer public keys in order to forge bids, or they could compromise smart appliances or transactive controllers in order to alter the bidding strategies of those devices. In contrast to attacking multiple prosumers individually, it may be much simpler to compromise a single node that is serving as a gateway for a group of prosumers. By way of illustration, the adversary can use vulnerabilities in the Ethereum software to either circumvent authentications or render miners inoperable. Here, we take into account three possible scenarios of an attack on miners, each featuring a distinct level of knowledge and sophistication on the part of the attacker.

One type of attack, known as a “gateway confidentiality and integrity attack”, occurs when an adversary compromises a gateway and gains access high enough to either hold off on recording certain bids or outright delete them. By reading both the bids submitted to the compromised gateway and the ones recorded on the blockchain by other gateways, the adversary can determine which bids to reject. The second type of attack, known as a “gateway integrity attack”, allows the adversary to delay or reject certain bids without compromising the network’s security. However, the attacker lacks full context and must make this decision based on limited information about the prosumers’ past bids. Third, an adversary can launch a DDoS attack against one of the gateways even though it cannot delay specific bids. As a result of this attack, the market cannot process certain bids, but the attacker is unable to read the bids as well.

### 4.2. Aim of the Opponent

An intelligent, self-interested foe is taken into account here. The adversary’s role determines its objectives and tactics (e.g., producer or consumer). In this paper, we examine the problem of prosumer bid rejection by adverse generators. Concretely, we assume that unfavorable generators seek market equilibria that boost the generator’s profit. In reality, IoT devices do not have the hardware or software to take part in the computationally intensive consensus algorithms used by many blockchains. As a result, prosumers can only connect to a blockchain-based system via gateway nodes, which an adversary can use to “cut off” prosumers from the system. For instance, an adversary can disrupt market equilibrium by launching a (distributed) denial-of-service (DDoS) attack against a gateway node and preventing a group of bids from reaching the market. As a result of this attack, the market cannot process certain bids, but the attacker is unable to read the bids as well. Because of the abovementioned reason, the authors do not consider DoS.

## 5. Implementation

This section outlines the deployment and implementation of the proposed framework. The prerequisites and setup environments are carried out as shown in Table 2.

### 5.1. Deployment of Nodes

The participants are defined in the *build.gradle* file. We created nodes such as a notary, producer, and consumer. The p2p ports were set as 1002, 1006, and 1006 for notary, producer, and consumer, as shown in Listing 1.

**Listing 1.** Deploying nodes on Corda network.

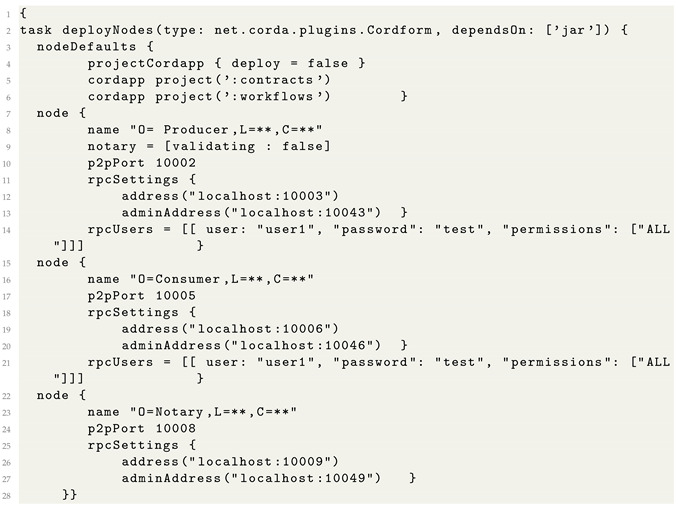



### 5.2. State Test

The energy contract and test files are designed with Java. A state test file is created using Java class. The producer, consumer, and matching notary and their identities are created using X500 certificates as shown in Listing 2. Here, we defined a method *energy state implements contract state ( )* and asserted the energy state with energy KW to be transferred, time of delivery, and matcher identities. The other rule we defined is that both the issuer and owner must be notified about any changes to the state. The third rule is to obtain the energy state details by a consumer such as price, KW, owner etc. Similarly, the energy state class is defined. Both the energy requester and owner are defined as participants. Then, the state is run to verify the three test cases defined previously.

### 5.3. Energy Trading Contract Test

Our network has three parties in it, i.e., producer, consumer, matching notary, and we defined them in the contract. To complete the contract test, we created two test cases for our request and transfer flow. For our transfer flow, we have an energy state, and for our transfer flow, we have one energy input state and one energy output state. The owners and traders are defined accordingly. The issue and transfer command test are also embedded here. The tests are defined with test cases such as contract with zero input in request transaction, with one output in request transaction, transaction output for energy state, and energy requester to be a required signer, as shown in the code. Similarly for transfer command, the test cases are designed. Here, CID represents the contract ID. The verify contract was also designed such that there is no null transaction and illegal argument exception.

**Listing 2.** Energy state test.

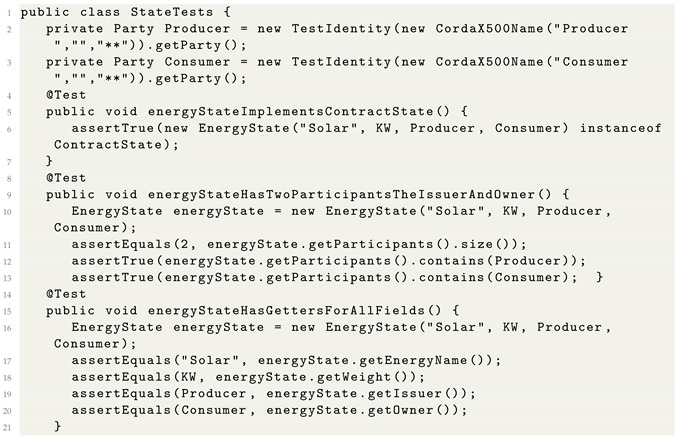



### 5.4. Running The Nodes

To run the Cordapp, we used *./gradlew clean deploy nodes* and *build/nodes/runnodes* in command prompt. After running the nodes, the energy request flow to producer and transfer flow to the consumer were performed. Command start request energy was used to run the flow. We used Ubuntu 18.04 with 16 GB of RAM. It took over a minute to start. The flow started by retrieving the notary, generating transaction, signing transaction with private key and sending flow to the counter party, obtaining notary signature, recording transaction, and broadcasting transaction to participate. To run the vault query, we obtained all the fields as output with time-stamping and transaction hash values.

### 5.5. Transaction Explorer

By running the transaction explorer from the command palates of the Corda extension for visual studio code, we chose producer, consumer, and notary to explore all the flow, such as request and transfer energy flow.Then, we filled other parameters such as energy type, demand and supply information in KW, and owner details to run the flow. The details of the transaction are depicted in Figure 4. It represents the timestamp, transaction hash, number of output, and other parameters. The details are shown in the vault (unconsumed states). Figure 5 depicts the vault query builder or view for producer and consumer, number of states, and transfer of energy transaction from producer to consumer.

## 6. Performance Analysis, Cyberattack Mitigation, and Discussion

In this section, performance metrics such as P2P, flow, metering, CPU and JVM heap memory, latency, and transaction rate (send and receive) are evaluated using visualization tools Prometheus and Grafana. In addition, mitigating a novel cyberattack, i.e., delay trading and discard, is discussed.

### 6.1. Mitigating Cyberattack

We mitigate the attack in our model by signing concept of notary, producer, and consumer, as shown in Equations (15)–(17). The demand and supply curve for adversary producer and consumer with an intention to delay bids was found. We analyzed that two supply curves intersect a single demand curve in the case of the producer delaying the bids. Out of two intersection points, one is normal and another one is an attack. In order to deal with a delay trading attack, we consider that if the consumer bid is delayed or if a consumer does not receive a confirmation, then it can resubmit its bid to notary node after waiting for a certain amount of time. In addition, the signatures associated with the producer (request and transfer type transactions), consumer (payment transaction), and notary (request, re-request, payment, repayment transactions) may reduce the effect of attacks by selecting the notary randomly. It creates an uncertainty for adversaries regarding which notary to target. Similarly, the double-spending attack is also tested by providing the same transaction multiple times, but the presence of the notary in the network prevents these transactions. It is found that this can also mitigate the double-spending attack.

### 6.2. JVM Heap Memory and CPU Performance

The summary of the performances measured for our proposed framework are illustrated in Table 3. In Table 3 the node name, flows started and completed, CPU and JVM memory usages, and flow duration are measured. Figure 6 and Figure 7 represent the CPU usage along with JVM memory usage at 15 min and 1 h of up-time. The Corda node maintains the number of caches. Basically, there are two types of cache in the Corda network, i.e., size- and weight-based. The producer, consumer, and notary JVM memory usage are presented. The notary takes more JVM memory usage compared to producer (party A) and consumer (party B). The CPU usage metric monitors the CPU load and overhead parameter of the network and returns an alert if high CPU measurements are found, as shown in Figure 8. The CPU usage is also shown in Figure 6 and Figure 7, where we found that the producer and consumer CPU usage is lower compared to the notary.

### 6.3. Measurement of P2P

The peer-to-peer metric is used to measure the messaging sequence between two parties, such as producer and consumer in our proposed application. It also measures the latency by calculating the number of messages sent and received between two parties. The size and interval of the sent and received messages are also obtained. Figure 9 represents the latency histogram of received and sent messages between energy participants. We found that after 15:30 (min:s) the receive and send latency between nodes is high, i.e., 115 ms and 105 ms, respectively, but it gradually decreases and the average measured are 22 ms and 51 ms for send and receive message latency.

### 6.4. Measurement of Message Rate and Flows

Figure 10 represents the message rate of send and receive transactions between nodes. The measured rate for received messages is 1.5 per second, whereas the send rate is 1.15 per second. The key activity among P2P energy trading nodes can be measured through flow metric. This metric includes the number of flows that started at a particular time, the number completed, and the number of flows that failed.

### 6.5. Metering

Metering metrics are used to measure the overall performance related to commands that are persistent, number of signature events, and waiting events queue length. We measured the transactions per second by varying single node with no notary (request) and two nodes with a notary (request and payment). We also found that the scalability varies with number of multiple payment transactions from one node to another via a notary.

### 6.6. Throughput

We compare the throughput by varying different types of flow such as energy request, request + pay, transfer, multiple notary, sender, receiver, and single notary. In the proposed framework, request is an energy asset that is based on payment state and contract in the P2P energy trading module, so in request flow, only one node with no notary appears on vault of the node. Energy request + pay flow interaction deals with two nodes, such as producer and consumer, to deal with request and transfer of asset ownership with the help of a notary. Request + repeated pay flow is requested on node A and repeatedly transfers a fraction of energy asset state to another node, B, through a notary. Finally, we found that spreading the transaction load over multiple notary clusters allows higher transaction throughput for the platform overall.

## 7. Conclusions

In this paper, we designed and deployed a blockchain-enabled peer-to-peer energy trading network in a Corda network with multiple peers and notaries. It addresses the issue of novel class of cyberattacks such as delay trading and discard. We also developed and analyzed various smart contracts, deployed nodes, state test, signer test, and transfer command contract rules in order to handle a novel peer-to-peer energy trading process in the proposed framework. The data privacy, availability, and security is maintained by using the notary service, transaction, and vault query services. The performances measured for the proposed framework with respect to CPU, JVM heap memory, transaction latency, throughput, message rate, and metering metrics were found to be optimum. The detailed implementation and prototype design was carried out with R3 Corda and IntelliJ tool for client communication. The framework provides an confidential identity security to all the trading participants by using network map services. The identities are only distributed to other participants on a need-to-know basis. We integrated use of the latest and far more reliable transaction broadcasting and validation services such as *notary*, *attachment*, and *network map* services. 

## Figures and Tables

**Figure 1 sensors-23-00670-f001:**
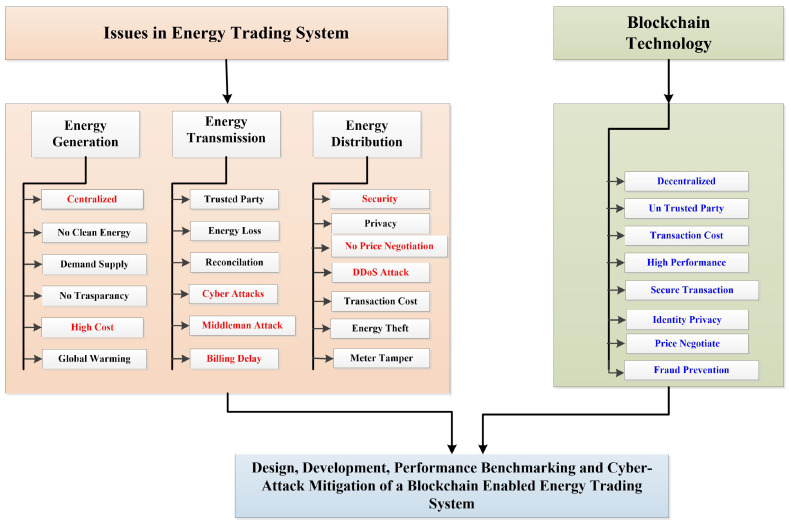
Issues in the traditional energy trading system.

**Figure 2 sensors-23-00670-f002:**
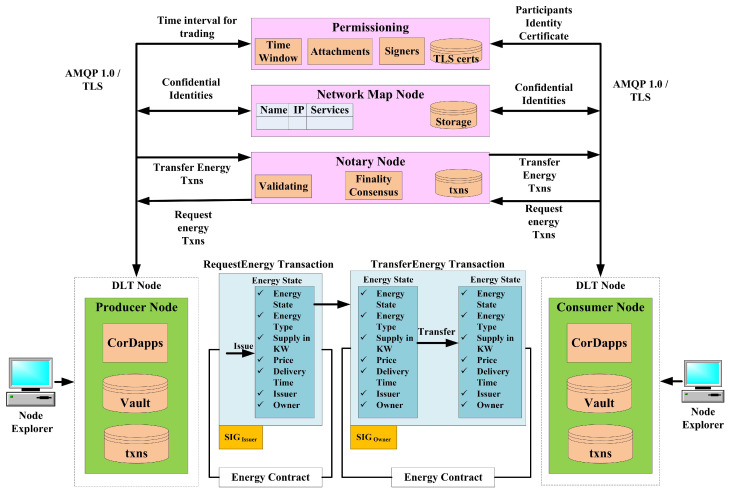
Proposed framework for P2P energy trading.

**Figure 3 sensors-23-00670-f003:**
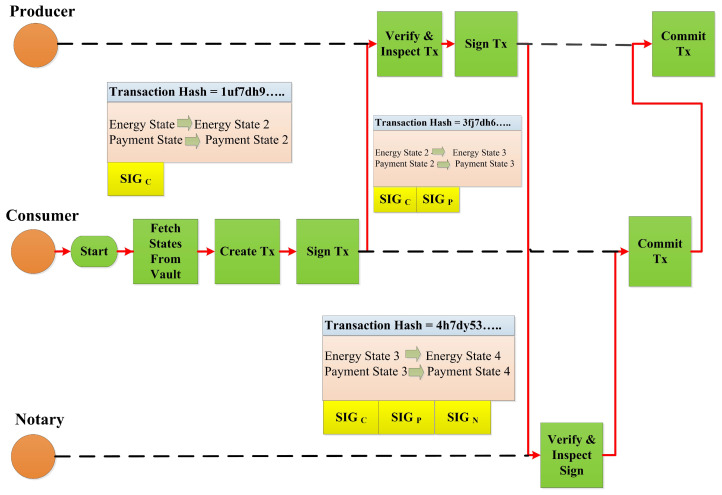
Energy request, transfer, and payment flow sequence in the proposed P2P energy trading framework with participant signatures.

**Figure 4 sensors-23-00670-f004:**
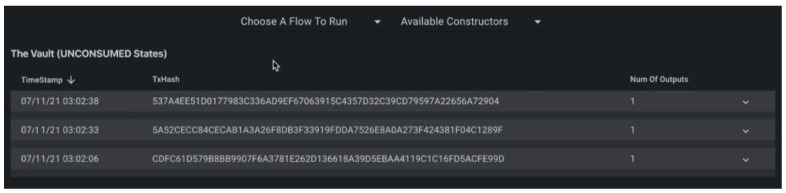
Transaction explorer.

**Figure 5 sensors-23-00670-f005:**
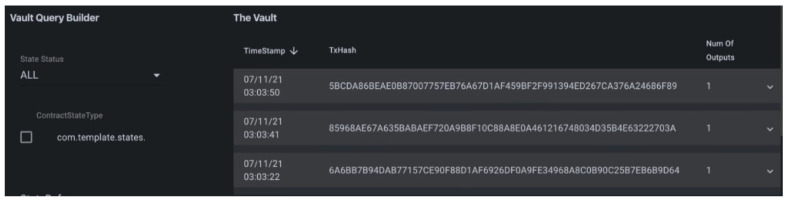
Vault query explorer.

**Figure 6 sensors-23-00670-f006:**
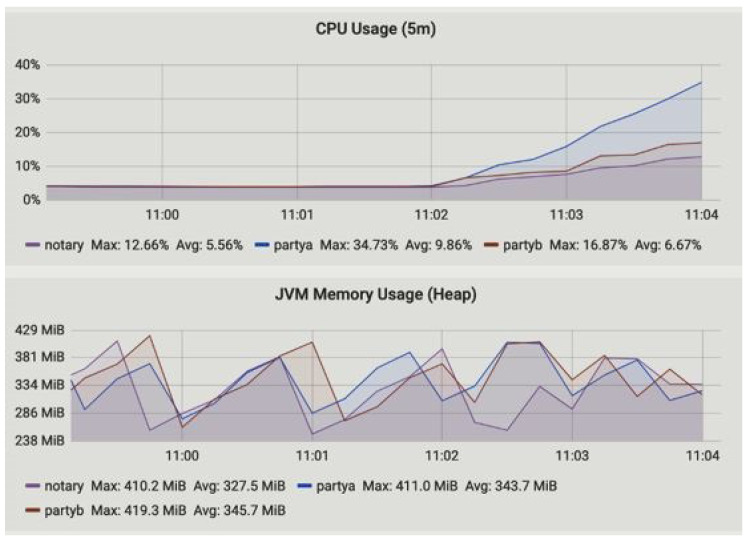
CPU and JVM heap memory usage at time t1 = 15 min.

**Figure 7 sensors-23-00670-f007:**
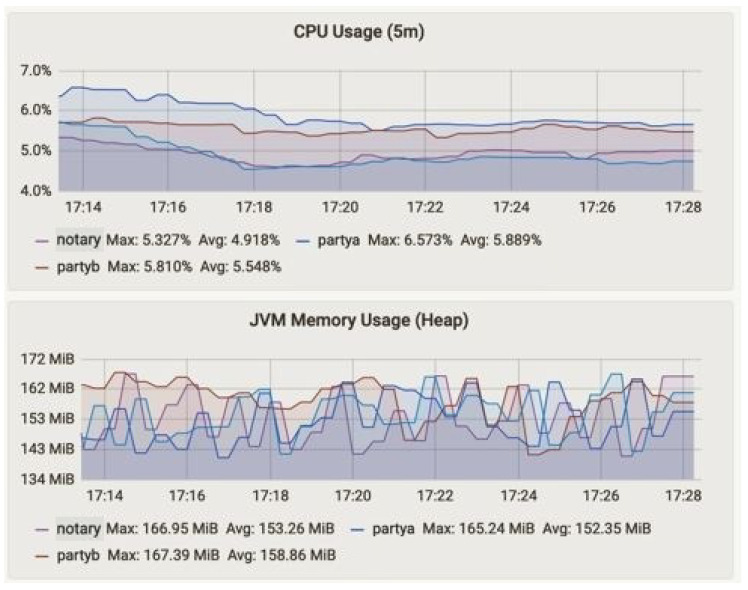
CPU and JVM heap memory usage at time t2 = 60 min.

**Figure 8 sensors-23-00670-f008:**
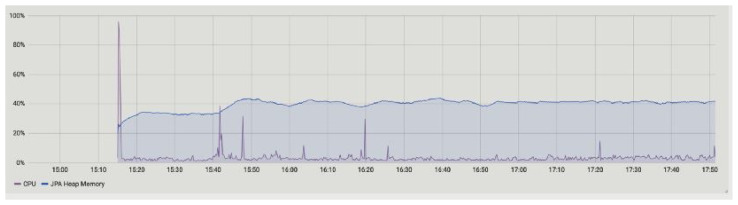
Overhaul CPU performance.

**Figure 9 sensors-23-00670-f009:**
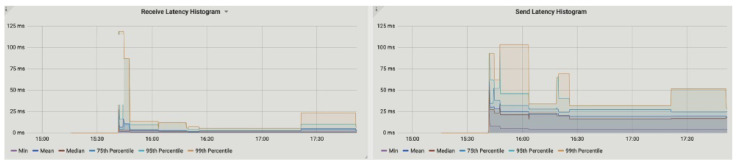
Receive and send latency histogram.

**Figure 10 sensors-23-00670-f010:**
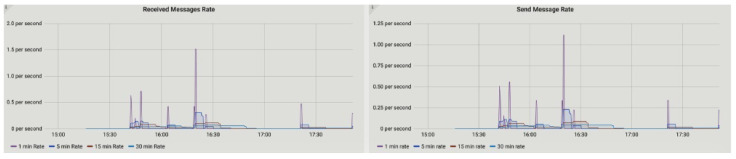
Send and receive message rate.

**Table 1 sensors-23-00670-t001:** Related work on blockchain-based peer-to-peer energy trading systems.

References	Year	Objective	Performance	Limitation	Performance Evaluation
Want et al. [19]	2019	Energy trading meets blockchain in electrical power system.	It slows down network computing over the time and makes the network less transparent.	Full transparency.	No
Chen et al. [20]	2018	Discussed vulnerabilities of load forecasting through adversarial attacks.	Only calculated average response time, throughput, and average message time based on Ethereum.	Full transparency.	Partially
Stellios et al. [22]	2018	To survey IoT-enabled cyberattacks and assessing attack paths.	Demonstrable different cyberattack in a blockchain network.	Increased computing power needed.	No
Pradhan et al. [23]	2022	IOTA, a lightweight ‘Tangle’-based framework (third-generation distributed ledger technology) to create a market for trading energy that uses a DAG.	Verifiable secure sharing of large number of microtransactions.	Only light wallet such as IOTA 2.5.4 IRI can support.	Partially
Pradhan et al. [1]	2021	This manuscript includes both an on-chain and off-chain permissioning scheme for energy users through the Orion and Metamask wallets.	SIt uses Hyperledger Besu and Istanbul Byzantine Fault Tolerant (IBFT) 2.0 consensus algorithm to implement contract.	Scalability.	No
Pang et al. [21]	2022	It gives a survey in detail on recent developments on the security of NCSs deception attacks.	Security incidents reported in recent years are reviewed and a couple of prevailing cyberattacks are analyzed.	Related to deception attack only.	No
Proposed Approach	2023	To design and propose an efficient blockchain-based peer-to-peer energy trading system with Corda services and cyberattack mitigation.	Transaction explorer, vault query explorer, CPU usages, JVM memory, flow started, flow stopped.	Fault tolerance.	Yes

**Table 2 sensors-23-00670-t002:** Software requirements and specifications for proposed Corda blockchain network.

Requirements	Specification
Operating system	Ubuntu Linux 18.04 (16 GB RAM) (64 bit)
Java	Java 8 and JDK
R3 Corda	VS Code-Extension Corda 0.0.3
Developer tool	IntelliJ IDEA 2021.3
cURL tool	Version 7.74.0
Docker engine	Version 17.06.2
Node JS	Version 10.21
NPM	Version 6.14.4
VS code	1.49.1
Grafana Prometheus Dashboard (performance measurement)	7.5.2

**Table 3 sensors-23-00670-t003:** Summary of the performances measured for our proposed Corda network.

Node Name	Flows Started	Flows Completed	CPU Usage		JVM Memory Usage (Heap)		Notary Flow Duration (5 m)	Uptime in Minutes
			Max.	Avg.	Max. in MB	Avg. in MB		
Producer	0	0	31.17	8.17	408.5	316.8	0	15
	1	1	10.79	4.58	408.5	321.0	1	30
	109	109	39.73	9.86	411.0	343.7	0.22	45
	404	400	5.81	5.54	167.39	158.86	0.8	60
Consumer	0	0	30.00	7.93	412.4	315.0	0	15
	1	1	8.39	4.13	415.1	312.3	1	30
	90	89	16.87	6.67	419.3	345.7	0.22	45
	410	416	5.71	4.89	166.87	154.11	0.8	60
Notary	0	0	28.97	7.64	406.5	312.3	0	15
	1	1	5.93	3.84	415.3	316.8	1	30
	96	96	12.66	5.56	410.2	327.5	0.22	45
	392	398	6.57	5.88	165.25	152.35	0.8	60

## Data Availability

The Corda code for implementing and verifying the presented blockchain-based peer-to-peer energy trading design is available in a publicly accessible GitHub repository. The prototype code can be found here: https://github.com/niharlipu13/p2pet-corda (accessed on 3 March 2022).

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
