# Peer review of "Performance Evaluation and Cyberattack Mitigation in a Blockchain-Enabled Peer-to-Peer Energy Trading Framework"

_sensors, 2023, doi:10.3390/s23020670_

Round 1
Reviewer 1 Report
I have the following two major concerns:
1. I would like to see a separate threat model section, where the adversarial capabilities are clearly laid out. Which kind of attacks are considered? Why DoS in not considered? Some elaboration on these would help improve the paper. Without proper assessment of the adversarial capabilities of the attacker, it is futile to judge the strength of a defense mechanism.
2. The authors did not provide any comparison with prior work. Is there no other work available in the domain? If not, please provide a comprehensive comparative analysis with prior work including performance and security.
Below are some minor comments:
1. There exists a lot of typos that can be fixed by running a spell-checker. This must be corrected before publication.
2. Algorithms must be presented in a better way, currently, it looks visually unsatisfactory.
3. There are too many listings provided that are superfluous to the paper. Please remove the unnecessary listings, if not required.
4. Some of the figures look really blurry, please use vector graphics format to include figures in the paper.
5. I would ask the authors to make the code available in github, if possible. This greatly improves the reproducibility of the work and enhances the knowledge of the community.
Author Response
Dear Editor-in-Chief,
MDPI Sensor
The authors thank the editor for getting our manuscript reviewed and giving us an opportunity to revise the manuscript entitled “Performance Evaluation and Cyber-Attack Mitigation in a Blockchain-Enabled Peer to Peer Energy Trading Framework” bearing Manuscript ID: sensors-2056343. Please accept the revised manuscript as a candidate for publication in the Journal: MDPI Sensors, Section: Internet of Things, Special Issue: Security, Privacy and Attack in Next Generation Networks.
We would like to sincerely thank the associate editor and reviewers for their time, effort, and thoughtful comments on the manuscript. They raise important issues and their inputs are very much helpful for improving the manuscript. We agree with almost all the comments and we have revised our manuscript accordingly. Modifications have been highlighted or written using different color in revised manuscript for easy identification. All figures have been reproduced in high quality (600 dpi). References have been set in format according to the journal guidelines. Moreover, English language and style have been revised in the full text.
Also a detailedresponse has been provided for each of the reviewer’s comments highlighting. In addition, we include how we have revised the manuscript based on those comments and recommendations. We hope that the reviewers will find our responses to their comments satisfactory. Please, find below the reviewer’s comments repeated in italics (highlighted in blue) and our responses inserted after each comment.
Sincerely,
K Hemanta Readdy, Ph.D.
Associate Professor
Department of Computer Science & Engg.
VIT University, AP, India.
Email:hemanth.reddy@vitap.ac.in,
Tel: +91---

Reviewer 2 Report
This paper designs and deploys a Blockchain enabled peer to peer energy trading network in a Corda network with multiple peer and notaries. The detailed implementation and prototype design is done with R3 Corda and IntelliJ tool for client communication. The performance parameters are analysed with Grafana visualization tool. Some comments about this paper are given as follows.
1. The Abstract and Keywords should be simplified.
2. In Section 1, the main novelty and contributions of this paper are not clear.
3. In Sections 1 and 2, more recent literature, for example, 10.1080/00207721.2022.2143735, should be reviewed.
4. Algorithm 2 and Table 2 are not clearly described.
5. In Section 4, Listing 1, 2, 3, 4, Figure 4 and Figure 5 may be deleted, and the implementation of each part is presented in words.
6. Some typos and grammar errors should be corrected. e.g., “Section 3 is about”, “Figure 8 and representsr”, and “and and”.
7. The presentation of this paper should be carefully polished, including equations such as (13) and (14).
Author Response
Dear Editor-in-Chief,
MDPI Sensor
The authors thank the editor for getting our manuscript reviewed and giving us an opportunity to revise the manuscript entitled “Performance Evaluation and Cyber-Attack Mitigation in a Blockchain Enabled Peer to Peer Energy Trading Framework” bearing Manuscript ID: sensors-2056343. Please accept the revised manuscript as a candidate for publication in the Journal: MDPI Sensors, Section: Internet of Things, Special Issue: Security, Privacy and Attack in Next Generation Networks.
We would like to sincerely thank the associate editor and reviewers for their time, effort, and thoughtful comments on the manuscript. They raise important issues and their inputs are very much helpful for improving the manuscript. We agree with almost all the comments and we have revised our manuscript accordingly. Modifications have been highlighted or written using different color in revised manuscript for easy identification. All figures have been reproduced in high quality (600 dpi). References have been set in format according to the journal guidelines. Moreover, English language and style have been revised in the full text.
Also a detailed response has been provided for each of the reviewer’s comments highlighting. In addition, we include how we have revised the manuscript based on those comments and recommendations. We hope that the reviewers will find our responses to their comments satisfactory. Please, find below the reviewer’s comments repeated in italics (highlighted in blue) and our responses inserted after each comment.
Sincerely,
K Hemanta Readdy, Ph.D.
Associate Professor
Department of Computer Science & Engg.
VIT University, AP, India.
Email:hemanth.reddy@vitap.ac.in,
Tel: +91---

Round 2
Reviewer 2 Report
This version is satisfying, and thus is acceptable.